# Contribution of an Artificial Intelligence Tool in the Detection of Incidental Pulmonary Embolism on Oncology Assessment Scans

**DOI:** 10.3390/life14111347

**Published:** 2024-10-22

**Authors:** Samy Ammari, Astrid Orfali Camez, Angela Ayobi, Sarah Quenet, Amir Zemmouri, El Mehdi Mniai, Yasmina Chaibi, Angelo Franciosini, Louis Clavel, François Bidault, Serge Muller, Nathalie Lassau, Corinne Balleyguier, Tarek Assi

**Affiliations:** 1Department of Radiology, Gustave Roussy, 94805 Villejuif, France; 2Laboratoire d’Imagerie Biomédicale Multimodale Paris-Saclay, Inserm, CNRS, CEA, BIOMAPS, UMR 1281, Université Paris-Saclay, 94800 Villejuif, France; 3Avicenna.AI, 375 Avenue du Mistral, 13600 La Ciotat, Franceyasmina.chaibi@avicenna.ia (Y.C.); angelo.franciosini@avicenna.ia (A.F.);; 4Division of International Patients Care, Gustave Roussy, 94805 Villejuif, France

**Keywords:** artificial intelligence, incidental, pulmonary embolism, oncology, assessment scans

## Abstract

Introduction: The incidence of venous thromboembolism is estimated to be around 3% of cancer patients. However, a majority of incidental pulmonary embolism (iPE) can be overlooked by radiologists in asymptomatic patients, performing CT scans for disease surveillance, which may significantly impact the patient’s health and management. Routine imaging in oncology is usually reviewed with delayed hours after the acquisition of images. Nevertheless, the advent of AI in radiology could reduce the risk of the diagnostic delay of iPE by an optimal triage immediately at the acquisition console. This study aimed to determine the accuracy rate of an AI algorithm (CINA-iPE) in detecting iPE and the duration until the management of cancer patients in our center, in addition to describing the characteristics of patients with a confirmed pulmonary embolism (PE). Materials and Methods: This is a retrospective analysis of the role of Avicenna’s CE-certified and FDA-cleared CINA-iPE algorithm in oncology patients treated at Gustave Roussy Cancer Campus. The results obtained from the AI algorithm were compared with the attending radiologist’s report and were analyzed by both a radiology resident and a senior radiologist. In case of any discordant results, the reason for this discrepancy was further investigated. The duration between the exact time of the CT scan and analysis was assessed, as well as the duration from the result’s report and the start of active management. Results: Out of 3047 patients, 104 alerts were detected for iPE (prevalence of 1.3%), while 2942 had negative findings. In total, 36 of the 104 patients had confirmed PE, while 68 alerts were false positives. Only one patient reported as negative by the AI tool was deemed to have a PE by the radiologist. The sensitivity and specificity of the AI model were 97.3% and 97.74%, while the PPV and NPV were 34.62% and 99.97%, respectively. Most causes of FP were artifacts (22 cases, 32.3%) and lymph nodes (11 cases, 16.2%). Seven patients experienced delayed diagnosis, requiring them to return to the ER for treatment after being sent home following their scan. The remaining patients received prompt care immediately after their testing, with a mean delay time of 8.13 h. Conclusions: The addition of an AI system for the detection of unsuspected PEs on chest CT scans in routine oncology care demonstrated a promising efficacy in comparison to human performance. Despite a low prevalence, the sensitivity and specificity of the AI tool reached 97.3% and 97.7%, respectively, with detection of all the reported clinical PEs, except one single case. This study describes the potential synergy between AI and radiologists for an optimal diagnosis of iPE in routine clinical cancer care. Clinical relevance statement: In the oncology field, iPEs are common, with an increased risk of morbidity when missed with a delayed diagnosis. With the assistance of a reliable AI tool, the radiologist can focus on the challenging analysis of oncology results while dealing with urgent diagnosis such as PE by sending the patient straight to the ER (Emergency Room) for prompt treatment.

## 1. Introduction

Cancer-associated pulmonary embolism (PE), constituting a substantial burden with increased cancer-related mortality, is dependent on the primary site of malignancy and the extent of the disease, but also on the characteristics of patients with significant comorbidities [1,2]. At autopsy, up to 12.4% of cancer patients exhibited one or more PE events, thus highlighting the high incidence of PE in this subset of patients [3]. In comparison to non-cancer patients, patients with malignant tumors have a 4–8-fold increase in the risk of mortality following an acute thrombotic event [4,5]. Moreover, the prevalence of venous thromboembolism events, including PE in cancer patients, is 12-fold higher than the general patients, but also more significant when receiving active anti-cancer therapy (23-fold) [6]. Therefore, a prompt diagnosis of PE is warranted in cancer patients to optimize their survival outcomes.

On the other hand, incidental pulmonary embolism (iPE) is commonly diagnosed during routine surveillance imaging, with an incidence varying between 1% and 5% [7]. Nevertheless, iPE is associated with respiratory symptoms that are mistakenly attributed to cancer itself. Therefore, iPE should not be considered a benign situation since it has been shown that the risk of recurrent events or PE-related events does not differ significantly from symptomatic PE. Chest computed tomography (CT) angiography remains the standard of care of imaging for the characterization of suspected PEs [8,9]. Current therapeutic strategies of iPE are similar to symptomatic PE, requiring the rapid instauration of active anticoagulation to prevent serious outcomes [7,10].

With the advent of artificial intelligence (AI) into daily medical activities, there is an unmet need to revolutionize and support the work of radiologists in the rapid diagnosis and management of acute medical conditions [11]. Improvement in the quality and technology of computed tomography (CT) scans has drastically changed the sensitivity in detecting PEs, with better visualization of the pulmonary artery and spatial resolution [12]. It is certain that AI, through the recognition of complex patterns in radiological data and the elaboration of a quantitative assessment of imaging features, would empower radiologists in their quest for enhanced accuracy and efficiency while minimizing the risks of diagnostic errors [13]. With the increasing number of FDA (Food and Drug Administration)-approved AI algorithms, radiologists and AI should become partners, not competitors, to enhance their performance and accuracy [14].

Therefore, numerous retrospective trials tried to assess the accuracy of AI-based models in detecting PEs [15,16,17,18,19]. AI improved the efficacy in the diagnostic process of suspected PEs; consequently, its implementation in daily routine care would enhance the detection of iPEs in populations like cancer patients [20]. In this study, with the aid of an AI-powered tool (CINA-PE Avicenna.AI, Avicenna.AI, La Ciotat, France), we hypothesized that the application of this model could support the radiologists in the diagnosis of iPE in an asymptomatic oncology population undergoing routine clinical CT scans to improve the clinical outcomes and reduce the time to the optimal therapeutic approach.

## 2. Methods and Materials

This study was approved by the Institutional Review Board of our hospital (no.: 2024-380). The need for written informed consent was waived. Avicenna.AI provided the PE-detection DL-based algorithm for this study. The study received no financial support. The authors had control of the data and the information submitted for publication at all times.

### 2.1. Study Design

The main objective of this study was to assess the role of AI in the accurate diagnosis of iPEs among cancer patients undergoing routine contrast-enhanced CT scans at Gustave Roussy Cancer Campus. The data were retrospectively and consecutively collected for a time period of 6 months from December 2022 to June 2023. All adult oncology patients who underwent a surveillance CT scan without suspected symptoms of PEs were included.

The patients included were being followed up for stage IV or metastatic neoplasia and had undergone an oncological evaluation CT scan with a thoraco–abdomino–pelvic CT scan with contrast.

Patients who had undergone a thoracic angioscan or thoracoabdominal angioscan and scans performed as part of an emergency or complication were excluded, as were initial assessments of tumors without histology.

Demographic data of patients and their disease, as well as information regarding the management of patients after detection of PEs, were also included.

The date and time of the CT exam (image acquisition) and the date the report was validated by the secretaries and medical assistants after the radiological report by the junior radiologist and correction by the senior radiologist was recorded.

The time and date of the patient’s admission to the emergency department for anticoagulant treatment was also recorded. Figure 1 shows a summary on the management with or without AI of incidentally discovered pulmonary embolism in cancer patients.

### 2.2. Imaging Feature and Analysis

#### 2.2.1. CT Scan Data

The scanner datasets were provided by a clinical center and were distributed appropriately in terms of suppliers (GE Healthcare GE Optima CT660, GE Healthcare, Milwaukee, WI, USA, which is a 64-slice CT scanner, Siemens Healthineers SOMATOM^®^ Force Dual Source CT system, Erlangen, Germany).

Two specialized radiologists (A.O. and S.A.) reviewed the positive results reported by the AI system, and data were filtered between true PEs and false positives. The specific causes of FPs were also detailed in the results section.

#### 2.2.2. AI System

The iPE deep learning (DL)-based algorithm consists of a series of Unets, a particular type of Convolutional Neural Networks (CNNs) trained to detect specific anatomical structures (refUNET). Unets, known for their effectiveness in handling biomedical images, implement an encoder–decoder framework. The encoder captures features at different levels of abstraction through a series of convolutional layers. The decoder then samples the abstract features to generate voxel-wise segmentation maps. This strategy allows the Unet to capture both high-level contextual information and fine-grained details of medical images, essential for accurate segmentation.

The detection of PEs is achieved by leveraging multiple stages, beginning with the preprocessing of medical images to enhance relevant features and normalize intensity levels. The preprocessed images are then fed into the Unet models, which generate voxel-wise segmentation maps of key anatomical landmarks and potential PE regions.

This DL-based tool was trained on 7630 CT scans and validated on 381 CT scans from 39 different CT scans models. The algorithm achieved a sensitivity of 87.8% (95% CI: 82.2–92.2%) and a specificity of 92.0% (87.3–95.4%).

### 2.3. Statistical Analysis

On a per patient level, sensitivity, specificity, positive predictive value (PPV), and negative predictive value (NPV) for the detection of PE were calculated. The 95% confidence intervals (95% CI) were calculated using the exact binomial distribution (Clopper–Pearson) (ref CP). To further investigate the potential impact on clinical workflow, the mean time between the exam and the physician’s interpretation was calculated. The number of patients that were discharged (patients that left the hospital) due to the missing PE diagnostics and then rehospitalized once the diagnosis was established was evaluated. Furthermore, we analyzed the number of patients that needed hospitalization and that initiated active anticoagulation treatment once the PE was detected. All the statistical analyses were performed using MedCalc Statistical Software (v20.015, MedCalc Software Ltd., Ostend, Belgium).

## 3. Results

A total of 3050 patients were included in the analysis, in whom the prevalence of iPEs was 1.3% (39 patients). The most common primary site of malignancy among patients with iPE was digestive (nine out of thirty-three with available clinical data; 27.27%), followed by thoracic (eight patients; 24.2%) and gynecological neoplasms (five patients; 15.2%) (Table 1).

In the first version of the model, out of 3049 evaluated patients, 233 alerts for suspicious iPE were detected, while 2816 tests were found to be negative. Out of the 233 patients, 39 patients had confirmed iPE on their routine CT scans, while 194 patients had false positive (FP) results. Therefore, the sensitivity and specificity of the AI model were 100% (95% CI: 90.97–100%) and 93.55% (95% CI: 92.62–94.41%), respectively, while the positive predictive value (PPV) and negative predictive value (NPV) reached 16.74% and 100%. In the FP analysis, the most common causes were an artifact (80 cases, 41.2%), an FP (36 cases, 18.6%), and lymph nodes (21 cases, 10.8%) (Table 2).

The mean time of delay between the exam and the interpretation by the physician to detect the PE reached 8.13 h, with a standard deviation of 15.48 h (95% CI: 3.21–13.05). Following the test interpretation, there was a delay in the initial management in 5 out of 32 patients (15.6%) (patients left the hospital after the tests). Five patients (15.6%) needed hospitalization following iPE detection for surveillance and instauration of therapeutic anticoagulation, while two patients (6.25%) had serious complications. Active anticoagulation with low molecular weight heparin was initiated in those with newly diagnosed PE (25 patients).

In an effort to improve the performance, the model was revised, using different datasets for training and testing, and a second set of data was performed. Two patients from the first set of patients were not included in the second analysis due to processing issues.

In the second set of analysis, out of 3047 patients, 104 alerts were detected for iPE, while 2942 had negative findings. In total, 36 of the 104 patients had confirmed PE, while 68 alerts were false negatives. In this set of analyses, only one patient reported as negative by CINA-iPE was deemed to have a PE by the radiologist. The sensitivity and specificity of the second version of the AI model were 97.3% (95% CI: 85.84–99.93%) and 97.74% (95% CI: 97.14–98.24%), while the PPV and NPV were 34.62% and 99.97%, respectively. In the FP analysis, the most common causes were an artifact (22 cases, 32.3%), an FP (17 cases, 25%), and lymph nodes (11 cases, 16.2%).

## 4. Discussion

The purpose of this paper is to assess the value of adding CINA-iPE, an AI system for the detection of unsuspected PEs on chest CT scans, in routine oncology care outside urgent medical conditions. With a reported prevalence of 1.3% of PEs, the sensitivity and specificity reached 97.3% and 97.7%, respectively. The AI tool detected all the reported clinical PEs except one case, while no cases of missed PEs were noted. These data, with two datasets that showed reduced FPs in the second dataset, support the fact that ML is a continuous process that would ultimately lead to an optimal AI system in support of radiologists.

An AI medical tool, to be considered a valuable diagnostic solution, must be fast and cheap with a significant impact on survival outcomes. Earlier data showed that AI tools used for the detection of intracranial hemorrhage (ICH) on non-contrast CT scans can reduce both image wait and turnaround times through ML algorithms, thus potentially improving the diagnosis and management of patients [21]. Therefore, with the knowledge that cancer patients have an increased risk of thrombotic events including PEs, there is an unmet need to optimize the diagnosis and treatment of PEs with the aid of AI. However, the vast majority of the published data assessed the role of AI tools in detecting PEs on CT pulmonary angiography (CTPA) in an emergency setting. In fact, a recent meta-analysis evaluated the actual role of DL in PEs on a total of 36,847 CTPA scans in seven studies, in a retrospective fashion, which led to a pooled sensitivity and specificity of 88% and 86%, respectively. With an acceptable rate of FPs, the DL model’s performance was considered satisfactory, despite the retrospective nature of these works [22]. Moreover, the implementation of a validated commercialized AI algorithm in 1202 patients with suspected PE has led to increased sensitivity (92.6%) and negative predictive value (NPV), thus increasing the self-confidence of radiologists in their final diagnosis [20]. This AI tool detected a number of missed diagnoses in the emergency setting which would have significant better outcomes in cancer patients [20] (Figure 2).

Nevertheless, in the oncology field, iPEs are common, with an increased risk of morbidity when missed with a delayed diagnosis, most particularly in patients undergoing multiple routine surveillance CT scans in both primary and secondary care. In a meta-analysis by Meyer et al., the occurrence of iPE among oncology patients reached 3.36% (963 out of 2826 screened patients), with the highest prevalence among prostate cancer patients followed by hepato-biliairy carcinoma and pancreatic tumors [23]. On the other hand, the sensitivity for PE diagnosis by the radiologists is commonly low, ranging between 67 and 87%, with a high specificity between 89 and 99% [24,25]. In specialized cancer centers, radiologists are commonly overwhelmed by the overflow of CT scans, thus highlighting the need for validated support in their routine daily activities. Several AI companies operating in the healthcare industry have gained FDA (Food and Drug Administration) clearances in various indications including iPEs. AIDOC medical, the only AI model used in the published retrospective trials for iPE detection, gained FDA clearance for triage of iPE in October 2020 [26]. In March 2024, Avicenna.AI, a medical imaging AI company, received clearance from the FDA forCINA-iPE, an AI tool for the detection of iPE on routine CT scans in different health conditions, thus reducing the risk of PE-related morbidity and mortality [27]. To the best of our knowledge, this is the first study evaluating CINA-iPE in the characterization of iPE in an oncology population undergoing routine surveillance imaging.

Several retrospective analyses with distinct populations and CT scan models were designed to address the dilemma of iPE in both the general and oncology populations [15,16,17,18,19] (Table 3). First, Wildman-Tobriner et al. implemented an AI model for reporting iPE in a total of 11,913 CT examinations in all comers, including oncology patients (both chest, abdominal, and pelvic (CAP) CT scans and abdominal and pelvic CT scans), and reported a rate of 0.47% and 0.34% of FNs on CAP scans and AP scans, respectively. Demonstrating a significant agreement with radiologists, this AI tool was capable of detecting 49 missed iPEs (0.41%), thus constituting around 38% of missed PEs by radiologists (49 out of 128 PEs) [18]. It is worth mentioning that assessing iPE in the general population might undermine the value of AI where the risk of thrombotic events is lower in comparison to oncology patients. Moreover, the high risk of missed PEs in this study in comparison to our paper (0%) highlights the difference in the qualifications of radiologists between centers, an issue that was not addressed in these different trials. In a similar fashion, Batra et al. performed a retrospective analysis on 2555 patients, with or without a cancer diagnosis, to detect iPE on conventional CT scans with AI (Briefcase for PE detection by AIDOC). The frequency of iPE reached 1.3% (40 patients), with seven cases only detected by the radiologists while four were only detected by AI, thus highlighting the importance of the synergistic work between physicians and AI. The specificity and sensitivity were estimated at 99.8% and 86.3%, both significantly lower than clinical results, and there were no subgroup differences in terms of Se and Sp despite a small sample size, but there was also no difference in terms of NPV and PPV. Reported false positives were attributed to lymph nodes and pulmonary venous filling defects, while FNs were due to altered anatomy [15]. More recent data from Langius-Wiffen and colleagues demonstrated that 2.2% of acute iPE (67 out of 3089 patients) with routine portal venous contrast-enhanced chest CT scans (37.1% of oncology patients) had an additional 25 missed cases (37.3%), thus increasing the sensitivity of the AI-tool (AIDOC Medical, Tel Aviv, Israel). However, the AI tool had a lower PPV due to a high rate of reported FPs, most commonly related to suboptimal pulmonary artery opacification [17]. We may expect that low PPV would be overcome in the future through fast adaptation and the deep learning of ML models.

Two other retrospective analyses were conducted on oncology patients. In an analysis by Wiklund et al. on 1069 cancer patients undergoing an elective CT, the prevalence of iPE was 4%, with a sensitivity and specificity reaching 90.7% and 99.8%, respectively, after the application of a DL cloud-based AI algorithm (AIDOC Medical) with a confirmed shorter turnaround time. A total of 59 initially missed PEs were detected, but it should be noted that all included CT images were reviewed by a radiologist, which does not reflect daily routine care [19]. In addition, Topff et al. applied the same AI-model on 6447 oncology patients, which found a prevalence of iPE of 1.2% in the overall population, and achieved a sensitivity and specificity of 91.6% and 99.7%, respectively. Interestingly the missed PE diagnosis dropped significantly from 44.8% to 2.6% with the AI algorithm [16]. The most common causes of FNs were a flow artifact (41.9%) followed by an abnormality adjacent to a pulmonary artery (29%) and technical artifact (22.6%). With a median process time of 3 min, almost all of the studies were available for analysis at the opening of the study, but the number of FPs reached 18.8%. An interesting finding worthy of debate in this trial is the detection of a majority of segmental and subsegmental PEs (83.3%), which can suggest an overdiagnosis of a medical condition with no impact on survival; but, nevertheless, data so far support an active anticoagulation in this setting [10,16]. Also, data analysis and retrieval might constitute a limitation for AI software (AIDOC Medical), with a risk of delaying the reporting of the results (7.3% of non-analyzed exams) [16]. The SAFE-SSPE is a non-inferiority trial testing clinical surveillance in comparison to active anticoagulation in patients with low-risk subsegmental PE, which would help physicians understand the value of reporting these missed PEs by the AI system [28].

Implementation of AI models has tremendous advantages in terms of increased diagnostic accuracy, but also represents several limitations which would implicate the necessity for a leading role for the radiologists to guide its optimal use in the clinical setting. One optimal approach was conducted by Vallee et al., who evaluated the value of adding a DL algorithm to the clinical performance of radiologists in the detection of PEs on CT angiographies, using the already validated AI tool, CINA-iPE. A total of 15.8% of PEs were detected, and the good classification performance was higher with the human–AI combination (*p* < 0.0001). Interestingly, the sensitivity of the combo increased to 92.5% in the detection of PEs in comparison to 81.7% when physicians worked alone, but also enhanced the diagnosis agreement between residents with better concordance [29]. This finding is a clear message that the current flow of trials should address the harmony of work between radiologists and AI models and not try to compare their performances alone. Therefore, we believe that our paper has several strengths. First, CINA-iPE successfully matched the clinical results reported by the radiologists with optimal agreement between both counterparts. Second, we have evaluated the clinical impact in those suffering from iPEs, which was also reported with the initiation of an optimal anticoagulation and the hospitalization of those suffering from severe consequences. Third, we highlight the wide difference in the prevalence of missed PEs between the different trials, thus confirming the major role of the human factor in the diagnosis of PE. Nevertheless, there are relative limitations to our study, such as a limited generalizability of the outcomes to the general population in the clinical context of cancer patients which should be carefully monitored (high risk of thrombotic events); however, some might consider it to be an advantage for AI for its implementation in the general population to be a matter of debate with the low incidence of PEs. The relatively low PPV translates the need for radiologists to rule out 2/3 of the cases flagged by the AI. We may be concerned by this additional workload for the radiologist, while we may expect an improved PPV through continuous training and optimization of the models. Another common challenge between all trials including ours is that some iPEs might be missed by both physicians and AI, since AI-negative findings are not routinely reviewed by physicians. Moreover, the low prevalence of iPEs might be attributed to the fact that all patients admitted to our hospital are offered prophylactic anticoagulation to reduce the occurrence of this complication in a vulnerable population.

That said, two clinical approaches can be proposed for an optimal alliance between AI and radiologists: first, AI can serve as a second reader to catch those PEs missed by radiologists, while the second strategy is that the radiologists review the red alarms detected by the AI tools to detect additional PEs at the expanse of missing the FNs of the AI system. Nevertheless, both options might pose a serious issue of time-consumption, given the high rate of FPs reported in numerous trials such as our paper. Therefore, a minimization of FPs reported by the AI algorithms is a must to increase their efficiency but also reduce the risk of burnout among physicians. These triage models can be extremely helpful in reducing the time to diagnose and treat PEs among patients undergoing CT scans during weekends, but also exams analyzed by teleradiology, a flourishing medical field mainly during the COVID period, therefore optimizing the quality of care [30]. With the absence of prospective studies in a clinical setting, there will be a need to assess the performance of these AI tools with a direct comparison of two clinical settings, the performance of radiologists with and without the aid of AI. In the near future, radiologists will have the privilege of time, with an emphasis on complex cases, while AI will take care of filtering normal CT scans.

In conclusion, this paper demonstrates the efficacy of CINA-iPE, an AI tool approved for detecting iPEs on chest CT scans in oncology patients with a high sensitivity and specificity, thus reducing the time for reporting positive findings. In the era of AI euphoria, the conception of an AI–radiologist alliance is a must for optimal outcomes in the healthcare field. In routine clinical practice, earlier detection of pulmonary embolism in oncology patients can enhance therapeutic strategies by reducing disease burden and enabling the timely initiation of treatment.

Our results demonstrate the good performance of this AI-based algorithm, opening the way for its use to help radiologists by highlighting the positive exams for PE in the worklist, thereby accelerating the diagnosis and communication workflow.

## Figures and Tables

**Figure 1 life-14-01347-f001:**
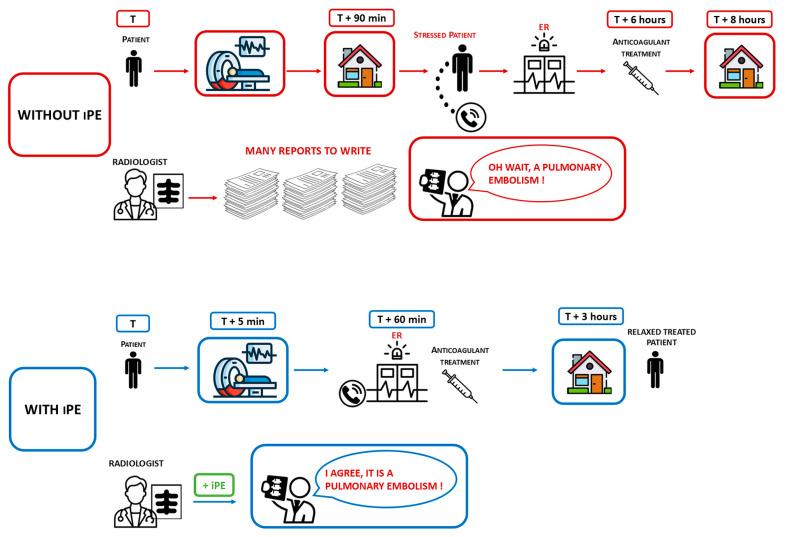
Summary of management with or without AI of incidentally discovered pulmonary embolism in cancer patients.

**Figure 2 life-14-01347-f002:**
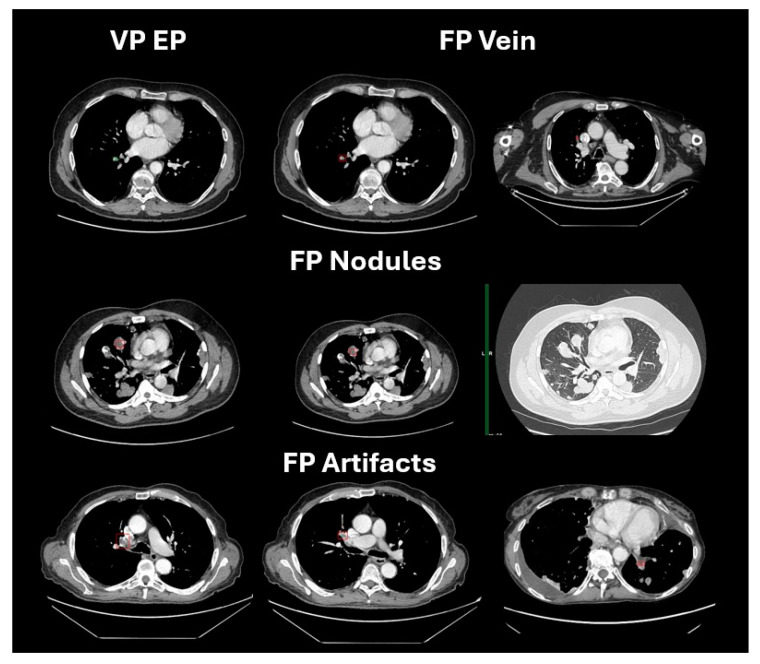
Algorithm results of CT scan images using CINA-iPE.

**Table 1 life-14-01347-t001:** Patient characteristics.

N	3050 Patients
Age (mean ± SD)	60.86 ± 12.44
Sex	
Male (%)	65.20%
Female (%)	34.80%
Characteristics of PEs	
Prevalence of PEs	1.3% (39 pts)
Primary Tumors (33 pts)	
Digestive	27.7% (9 pts)
Thoracix	24.2% (8 pts)
Gynecological	15.2% (5 pts)
Urological	9.1% (3 pts)
Head and Neck	9.1% (3 pts)
Breast	9.1% (3 pts)
Others	6 pts (2 pts)
Time between interpretation and exam	
Mean ± SD	8.13 ± 15.48
95% CI	[3.21–13.05]

**Table 2 life-14-01347-t002:** Overall results of the two versions of the dataset.

		CONFUSION MATRIX Version 1			CONFUSION MATRIX Version 2		
		CINA-iPE			CINA-iPE		
		iPE	NOT iPE	ALL	iPE	NOT iPE	ALL
GT	iPE	39	0	39	36	1	37
	NOT iPE	194	2816	3010	68	2942	3010
	ALL	233	2816	3049	104	2943	3047
Sensitivity		100.0%			97.3%		
Specifcity		93.6%			97.7%		
Accuracy		93.6%			97.7%		
PPV		16.7%			34.6%		
NPV		100.0%			100.0%		

**Table 3 life-14-01347-t003:** Published studies on AI detection of iPE.

Author	AI Model	Number of Patients	Number of CT Scans	Population	iPE Prevalence (%)	Se	Sp	PPV	NPV	Missed PEs (%)
Batra et al (2022) [15]	AIDOC	2555	3003	All comers	1.3%	82.5%	99.8%	86.8%	99.8%	10% (4 PEs)
Topff et al. (2023) [16]	AIDOC	6447	11,736	Cancer pts	1.3%	91.6%	99.7%	80.9%	99.9%	44.8% (47 PEs)
Langius-Wiffen et al. (2023) [17]	AIDOC	3089	3089	All comers	2.2%	95.5%	99.6%	85.3%	99.9%	37.3% (25 PEs)
Wildman-Tobriner et al. (2021) [18]	AIDOC	4087 (CAP)–4779 (AP)	11,913	All comers	0.66%	62%/ 61.2%	99.97%/ 99.98%	96.1%/ 96.8%	99.5%/ 99.7%	38% (49 PEs)
Wiklund et al. (2022) [19]	AIDOC	1004	1892	Cancer pts	4%	90.7%	99.8%	95.6%	99.6%	81.5% (53 PEs)
Our Study (2024)	CINA-IPE	3049	3049	Cancer pts	1.3%	97.3%	97.74%	34.62%	99.97%	

## Data Availability

The original contributions presented in the study are included in the article, further inquiries can be directed to the corresponding author.

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
