# Peer review of "Contribution of an Artificial Intelligence Tool in the Detection of Incidental Pulmonary Embolism on Oncology Assessment Scans"

_life, 2024, doi:10.3390/life14111347_

Round 1
Reviewer 1 Report
Comments and Suggestions for Authors
The study is highly contemporary, illustrating the significance of artificial intelligence in today's context, which can work as partners to physicians rather than competitors in the diagnosis and management of patients with pulmonary embolism. The text is well-written and clear.
Author Response
The authors would like to thank the editor and reviewers for their insightful and constructive comments. The authors hope that they have been able to satisfactorily address all of the reviewers’ concerns.
Reviewer 1:
Thank you very much for your comments.
Reviewer 2 Report
Comments and Suggestions for Authors
The article explores the use of an artificial intelligence (AI) tool, CINA-iPE, for detecting incidental pulmonary embolism (iPE) in oncology patients undergoing routine CT scans. The study examines the effectiveness and accuracy of this AI algorithm compared to traditional radiologist assessments.
Comments
1. provide the detailed discussion
2. highlight the key results on patients care
Author Response
Response to reviewers
The authors would like to thank the editor and reviewers for their insightful and constructive comments. The authors hope that they have been able to satisfactorily address all of the reviewers’ concerns.
Thank you very much for your comments.
- Provide a detailed discussion
- Thank you for your suggestion. We made several changes in the discussion (changes made are highlighted).
- Highlight the key results of patient care
- Thank you. We added a citation in the conclusion to highlight the results on patient care. Other key results are highlighted in the discussion.
Reviewer 3 Report
Comments and Suggestions for AuthorsТhe proposed paper is devoted to the description of a new approach based on artificial intelligence applied for detection of incidental pulmonary embolism and assessment of chest computed tomography scans.
Preliminaries to the research area are provided, mentioning the role of the artificial intelligence and computed tomography scans in the rapid diagnosis and management of acute medical conditions. The characteristics of cancer-associated pulmonary embolism are described and its influence on the increase of cancer-related mortality is mentioned. Recent retrospective trials tried to assess the accuracy of AI-based models in detecting pulmonary embolisms are reviewed.
The proposed approach is described in detail. Information about the study design and methods of imaging feature and analysis is provided.
The obtained results are described illustrated by many figures and tables. A detailed discussion is provided. The advantages and limitations of the proposed methodology are discussed.
The presentation of the main results is clear and comprehensive. The results are valuable and worthy of being published considering their possible applications in cancer research improving the efficiency and effectiveness of the detection of unsuspected pulmonary embolism on chest computed tomography scans.
Minor revisions are suggested to improve the quality of the exposition:
My main remark concerns the bad formatting of the manuscript. For example:
p. 1, lines 4-5: The names of two of the authors begin with small letter.
p. 2, line 53: The meaning of the abbreviation should be explained; please check also other abbreviations.
In many places large intervals within the text can be found.
The numeration of Subsection “AI system” is incorrect.
Captions and citation of many tables and figures are incorrect.
There are many commands like “cite” within the sentences.
Some of the figures should be moved slightly to the left.
p. 6, lines 199 - 200: There is a mistake – the word “sensitivity” is repeated incorrectly.
Months in the data for the References should be written in English.
Comments on the Quality of English LanguagePlease check for typos.
Author Response
Response to reviewers
The authors would like to thank the editor and reviewers for their insightful and constructive comments. The authors hope that they have been able to satisfactorily address all of the reviewers’ concerns.
Thank you very much for your comments.
- The names of the authors were modified as well as the format of the affiliations.
- Abbreviations were corrected.
- Intervals were removed.
- Numeration of the AI system was corrected.
- The commands such as cite were corrected.
- The word sensitivity was replaced by specificity.
- Months in references were corrected.
Reviewer 4 Report
Comments and Suggestions for Authors
authors report results of an AI to detect asymptomatic PE in oncology.
the strategy may be useful in the cinical practice but the contest has not fully investigated : stadation of investigated oncological patients should be reported in order to undertsand in which oncological setting diagnosis of PE may be useful.
furthermore, advantage for clinical practice and treatment strategy to improve the outcome of reported patients should be a milestone of their conclusions and it needs to be better underlined in the text
Author Response
-
Response to reviewers
The authors would like to thank the editor and reviewers for their insightful and constructive comments. The authors hope that they have been able to satisfactorily address all of the reviewers’ concerns.
-
Stadation of investigated oncological patients should be reported in order to understand in which oncological setting diagnosis of PE may be useful.- Thank you. We agree that stratification of patients might be of benefit to the study. The purpose of this study is to address all oncology patients with different stages as a screening tool for incidental PE. As reported in our paper, the retrospective nature of this study might be a limitation and prospective trials might be needed for a better understanding of the oncological setting of diagnosing PE.
- The advantage of clinical practice and treatment strategy …
- Thank you very much for your advice. The following citation was added and is highlighted in text: “In routine clinical practice, earlier detection of pulmonary embolism in oncology patients can enhance therapeutic strategies by reducing disease burden and enabling timely initiation of treatment.”
Additional modifications:
- We added a phrase to indicate that the second version of the algorithm was trained and tested on independent datasets different from the ones of the first version.
Round 2
Reviewer 4 Report
Comments and Suggestions for Authors.